# Design of Broadband High-Gain Fabry–Pérot Antenna Using Frequency-Selective Surface

**DOI:** 10.3390/s22249698

**Published:** 2022-12-11

**Authors:** Xianjun Sheng, Xiaolong Lu, Ning Liu, Yunhong Liu

**Affiliations:** School of Electrical Engineering, Dalian University of Technology, Dalian 116024, China

**Keywords:** frequency-selective surface, Fabry–Pérot resonator antenna, broadband, high gain

## Abstract

In this paper, a broadband high-gain Fabry–Pérot (F-P) antenna composed of the air-loaded slot-coupled broadband microstrip antenna and the frequency selective surface (FSS) based positive gradient reflection phase structure is proposed. Taking advantage of the superposition effect of multiple reflections and transmissions occurring between layer structures, the gain enhancement was realized. Meanwhile, by cascading the single-layer FSS and the dielectric substrate, the positive gradient reflection phase over a wider frequency range was achieved. Simulated results show that the resonant frequency of the designed F-P antenna is 10 GHz, the impedance matching band (S11 < −10 dB) ranges from 8.3 GHz to 11.25 GHz with a bandwidth of 29.5%, and the antenna gain is improved significantly in the range of 8.1 GHz~11.25 GHz with a gain bandwidth of 31.5%. For further verification, a prototype was fabricated, and the experimental and simulated results are in good agreement.

## 1. Introduction

Broadband and high-gain antennas play an extremely important role in wireless communications, such as satellite communications, remote control, and navigation [1]. Generally, the improvement of antenna gain can be realized by two methods: antenna structure optimization or antenna array structure design. However, applying the aforementioned method to improve the antenna gain often leads to problems such as complex antenna structure, increased processing cost, enhanced mutual coupling between antennas, and difficulty in designing a feeding network. Therefore, overcoming these problems by designing an antenna with a simple structure on the basis of high antenna gain and wide bandwidth is highly desired in practical applications.

Fabry–Pérot (F-P) antenna is a high-gain, high-directional antenna, which can avoid the disadvantages of traditional high-gain antennas, such as complex structure, large size, and high cost, and can overcome the large mutual coupling and feeding between array antennas. It is an effective solution for high-gain antenna design. However, the operating bandwidth of traditional F-P antennas is relatively narrow, which makes it difficult to meet the needs of high-gain antennas for broadband communication. In order to increase the bandwidth of the F-P resonator antenna, different methods and structures have been proposed [2,3,4,5,6,7,8,9,10,11,12,13]. For example, a partially reflective surface (PRS) with a positive slope of the reflection phase frequency response curve is adopted to construct the electromagnetic bandgap structure (EBG) resonant cavity antenna with an impedance bandwidth of 12.6% and a gain bandwidth of 15.7% in [6]. A dual-layer FSS structure is designed to construct the cladding layer of the antenna in [7], which extends the gain bandwidth and improves the gain simultaneously. A method of adding dielectric cladding to sparse arrays to expand the gain bandwidth of the antenna is presented in [8]. A double-layer dielectric plate is used as the antenna cladding to improve the gain of the antenna and increase its gain bandwidth to 27% in [5]. A coplanar waveguide wideband strawberry artistic-shaped printed monopole (SAPM) antenna is proposed in [9], and a monolayer frequency-selective surface (FSS) is used as a metal plate to improve the gain of antenna application. The proposed FSS reflector uses a 10*10 array, and the study uses a common surface waveguide (CPW)-fed FR4 substrate to print the proposed antenna, which provides a wide impedance bandwidth of 8.85 GHz (3.05 GHz–11.9 GHz), covering the licensed broadband. Due to its relatively high reflectivity size and positive phase gradient, the PRS layer proposed in [10] is a promising super velocity for wideband high-gain F-P antenna, which effectively improves the gain and bandwidth of array antennas. A wideband high-gain rectangular microstrip array antenna with a new frequency-selective surface (FSS) designed as a reflector for applications below 5 G is presented in [11]. The antenna configuration consists of a 1*4 rectangular microarray array antenna and FSS reflector to generate a semistable high radiation gain. In [12], the perforated metal layer is used as a frequency-selective surface (FSS) for printing groove dipole antennas operating in the V-band. The 3 dB gain bandwidth is from 61.1 GHz to 64.9 GHz (6.03%). The gain and bandwidth of the F-P antenna were increased by using a double-layer FSS in [13], resulting in an antenna gain of 16.8 dBi, impedance bandwidth of 18.4%, and gain bandwidth of 12.5%. These above studies show that the gain bandwidth of the F-P antenna can be effectively improved by rationally designing the cladding structure of the F-P antenna. Although these aforementioned research studies provide a solution for the broadband high-gain F-P antenna, the relative gain bandwidth of the proposed antenna structure still needs to be further improved to meet the needs of broadband communication, and it is urgent to design an F-P antenna with a wider gain bandwidth.

In this paper, an F-P antenna with excellent gain is designed. The antenna uses a cascade structure of a single-layer FSS and a dielectric layer as the antenna cladding, which can achieve positive gradient reflection with a higher slope in a wider frequency range phase. By loading it on the air-loaded slot-coupled broadband microstrip antenna, the antenna gain can be significantly increased in the frequency range of 8.1 GHz~11.25 GHz, with a maximum in-band gain of 10.35 dBi and a gain bandwidth of 31.5%.

## 2. Theoretical Analysis and Unit Design

### 2.1. Fabry–Pérot Resonant Cavity Theory

As shown in Figure 1, a conventional FP resonator antenna is formed by placing an EBG structure as a partially reflective surface (PRS) at a proper distance from the ground plane, which creates an air-filled cavity between the PRS and the ground plane, and fed by a small antenna or an array [7]. The main characteristics of the antenna, such as its operating frequency, directivity, gain bandwidth, and radiation patterns, are determined by the property of the PRS. Part of the electromagnetic waves radiated by the antenna passes through the coating directly, and the other part is reflected once or several times before passing through the coating. The two electromagnetic waves are superimposed in phase to achieve the effect of improving the antenna gain.

According to the ray theory, the frequency, directivity, and gain bandwidth of the antenna are affected by the characteristics of the F-P resonator and the loading height. When (1) is satisfied, the directivity coefficient of the antenna reaches the maximum [7].
(1)$$h=\frac{c}{4\pi f}+(\varphi_{PRS}+\varphi_{GND}-2N\pi ),N=0,1,2\cdots$$
where *h* is the depth of the cavity, *f* is the resonant frequency of the antenna, *c* is the velocity of light, *N* is the order of the resonance mode, *φ_PRS_* is the reflection phase of the cladding structure, and *φ_GND_* is the reflection phase of the ground plane. Normally, *φ_GND_* is *π*, and *φ_PRS_* can be derived from (1) as follows:(2)$$\varphi_{PRS}=\frac{4\pi h}{c}f+(2N-1)\pi ,N=0,1,2\cdots$$

As indicated by (2), once the reflection phase of the cladding structure is positively correlated to the frequency with a reflection slope of 4*πh*/*c*, the working band of the F-P antenna is broadened. Thus, obtaining the reflected phase indicated by (2) in a wide frequency range is important for designing wideband high-gain F-P antennas.

### 2.2. Selection of Feed Antenna

The traditional microstrip patch antenna has a narrow bandwidth and is not suitable for bandwidth expansion of F-P antennas, while the air-loaded slot-coupled microstrip antennas have a wider bandwidth and are suitable for high-gain antennas [5]. The structure of the microstrip slot antenna used is shown in Figure 2. The feed antenna adopts a rectangular radiation patch and has rectangular grooves in the floor plate. Multiplicative resonances are generated by coupling between radiating patches, rectangular slots, and microstrip lines, which result in a broad impedance bandwidth. Commercial laminate Rogers RT/duroid 5880 (*ε_r_* = 2.2, tan *δ* = 0.0009) with a thickness of 1 mm is used in the design.

This patch is coupled to the feeder through a ground plane gap, and the patch and ground plane are separated by an air gap to suppress surface waves, which can be fed into the cavity and degrade the antenna performance of the F-P resonator [14]. The antenna is simulated by HFSS15.0, and the parameters of the coupling feed of microstrip slots are determined and shown in Table 1.

The simulated reflection coefficient results of the microstrip slot-coupled feed antenna are shown in Figure 3. It can be seen that the reflection coefficient stays below −10 dB in the frequency range of 8.8 GHz to 11.4 GHz with an impedance bandwidth of over 20%. Obviously, compared with the traditional patch antenna, the antenna gain is improved (shown in Figure 4). However, it is still not suitable for high-gain applications. Considering the good impedance match and antenna gain over a wide frequency band, the air-loaded slot-coupled antenna is a good candidate for the feed source of the broadband F-P antenna.

### 2.3. Design of Positive Gradient Reflection FSS Unit

The traditional F-P antenna has narrow frequency bands because of the negative correlation between the reflection phase and cladding structure frequency. In order to expand the antenna bandwidth (mainly gain bandwidth), an FSS with a positive gradient reflection phase is designed (reflection gradient phase is the change in reflection phase that increases or decreases with frequency).

The traditional cladding has a relatively narrow bandwidth. For bandwidth broadening, this paper selects the FSS structure as the cladding structure. At low frequencies, the metal square patch FSS has a small reflection coefficient, and with the increase in frequency, the reflection coefficient increases. It has the characteristics of a low-pass filter. However, the wire grid has a contrary trend. The reflection coefficient is higher at low frequencies and decreases with increasing frequency. It has the characteristics of a high-pass filter. Therefore, the FSS structure combined with these two structures produces a weak resonance in the wide band, which results in a positive slope and high reflection coefficient of the reflection phase frequency response curve in this band.

The design of the FSS consists mainly of two parts, as shown in Figure 5. The upper layer is a pure dielectric plate, and the lower layer is a double-layered FSS structure. In the double-layered FSS structure, the upper FSS array is composed of the square slot and the bottom array is the wire grid. The two-layer dielectric board is made of Rogers RT/duroid 5880 (*ε_r_* = 2.2, tan *δ* = 0.0009), and the thickness is 1 mm.

The PRS is simulated by HFSS15.0, and the reflection phase of the PRS together with the ideal reflection phase indicated by (2) is plotted in Figure 6. In addition, the reflection phase of the FSS in the proposed PRS is also plotted in Figure 6, and it can be seen that if only the FSS in the proposed PRS is used, the frequency range with the normal reflection gradient phase is about 1 GHz. However, when the proposed PRS is used, the frequency range of the regular reflection phase gradient increases to 2 GHz (that is, with a positive reflection gradient phase in a wider frequency range), and a high reflection slope can also be observed in the designed PRS structure as shown in Figure 6.

There is a coupling effect between the FSS and the dielectric plate, which depends on the air height *h_c_*, as shown in Figure 7. This behavior of PRS is similar to the Fano-like resonance, which arises from the interference between nonradiative and radiative modes [15] and in different dielectric metal composite metamaterials [16,17] and nanophotonic structures [18]. The small air gap between the two PRSs has a stronger effect on the reflection properties of the multilayer PRS, as shown in Figure 8. As the band gap increases, however, the effect of the Fano resonance decreases due to the weaker EM coupling between the two PRSs.

The amplitude and phase of the reflection coefficient depend on the size of the FSS unit, and only a chosen size is suitable for designing an ideal FSS. It can be seen from Figure 5 that there are four parameters in the FSS unit that can be modified and improved. The F-P antenna will have strong broadband operating characteristics while keeping high gain only if the parameters take a moderate value, preventing the reflection amplitude of the FSS unit from being too low and the reflection phase curve from being too steep. The specific structural parameters are shown in Table 2.

The equivalent circuit diagram of the PRS unit is shown in Figure 8. The upper dielectric plate of the unit structure is *Z*_1,_ and the middle dielectric plate is *Z_air_*. *Z*_2_ is the dielectric plate of the lower FSS layer, *C_p_* and *L_p_* are the equivalent parameters of the upper FSS layer, and *L_S_* is the equivalent parameter of the lower FSS layer. It is known that the equivalent circuit is mainly affected by the structure change of the FSS layer.

The final result was obtained by HFSS15.0 simulation. The positive phase gradient obtained can cover a certain working frequency band while ensuring a higher slope, and the result is presented in Figure 9. As shown in Figure 10, the magnitude of the reflection coefficient is greater than 0.5, and the reflection coefficient of the entire frequency band is large; the transmission coefficient is greater than 0.5, and the transmissibility within the band is better. Figure 9 shows that the reflection phase at the center frequency of 10 GHz is 162°, the reflection phase curve with a positive slope is obtained in the frequency band of 8.9 GHz~11.3 GHz, and it can meet the requirements of antenna gain broadening in a wide frequency domain.

A PRS with a double-layer pure dielectric layer is proposed in [5], but it requires that the two dielectric layers be selected with different dielectric constants, the dielectric constant of the lower dielectric layer is greater than that of the upper layer, the distance between the two dielectric layers is greater than *λ*/2, and a PRS with a positive gradient reflection phase can be obtained, which limits its application. In this paper, by connecting the FSS layer with the dielectric layer, the FSS layer can be made of the same material as the dielectric layer, and the distance between the two is small. A PRS with a positive gradient reflection phase is easier to achieve through the coupling between the FSS layer and dielectric layer.

### 2.4. Design of F-P Resonant Cavity Antenna

A high-gain wideband F-P antenna is designed as a combination of the superstructure and feed antenna. According to the configuration shown in Figure 11, there are still two parameters, *W* and *h,* that are not determined.

Assuming that the lateral size *W* is infinite, the directivity of the F-P antenna relative to that of the feed antenna can be formulated as:(3)$$D_{r}=10\log\frac{1+\Gamma }{1-\Gamma }$$
where *D_r_* is the relative directivity and Г is the reflection magnitude of the superstrate structure. Ludovie et al. found that the electromagnetic field in the resonator can be equivalent to a Gaussian distribution [19] and then concluded the empirical formula for the radiation area *S* required by the resonator antenna to reach a certain gain value:(4)$$S=\frac{10^{D/10}\cdot \lambda^{2}}{0.8\pi^{2}}$$
where *S* is the antenna surface area, *D* is the total gain value added by the feed antenna gain and the gain value added by the PRS, *λ* is the free space wavelength at the operating frequency, and 0.8 is the effective power factor summed up after numerous simulation experiments. As can be seen from Figure 4, the gain of the 10 GHz feed antenna is about 7.5 dBi, and the reflection coefficient amplitude of the cladding FSS is 0.5. Therefore, the gain of the F-P antenna at 10 GHz is estimated at 12 dBi. Considering that broadband is required in practical work, W should be slightly larger. The initial determination of *h* can be based on (1), and further optimization needs to be performed through full-wave simulation.

As shown in Figure 12, studies have shown that *h* has a significant effect on F-P antennas. As *h* increases, the impedance bandwidth increases. For example, when *h* = 12.5 mm, the input impedance matches the wave impedance of the free-space band, resulting in a larger reflection (S11 > −10 dB). When *h* increases to 17.5 mm, the impedance matches well, and the impedance bandwidth reaches 32%.

Meanwhile, the gain bandwidth of the F-P antenna decreases with h. For example, at *h* = 12.5 mm, the gain bandwidth reaches 33%, but at *h* = 17.5 mm, the gain bandwidth is only 8%. In practical applications, the antenna impedance bandwidth and gain bandwidth are required simultaneously.

Hence, the value of h should be selected properly for the simultaneous enhancement of the impedance bandwidth and gain bandwidth. In our work, *h* is set to be 14.5 mm and the antenna has a better impedance match with a 23% impedance bandwidth and 33% gain bandwidth.

The side view of the broadband F-P antenna is shown in Figure 11. It was modeled and simulated by HFSS15.0. Through optimization, we can finally obtain *W* = 70 mm and *h* = 14.5 mm (about *λ*/2).

As shown in Figure 13, the number of FSS unit cells has a great impact on the performance of F-P antennas. As the number of FSS unit cells increases, the antenna’s impedance bandwidth increases first and then decreases. For example, the impedance bandwidth is 18% when the number of cells is 5*5, 22% when the number of FSS cells is 7*7, and 20% when the number of cells is 9*9. As the number of cells increases, the antenna gain bandwidth increases first and then decreases. For example, the gain bandwidth is 30% when the number of cells is 5*5, 33% when the number of cells is 7*7, and 26% when the number of cells is 9*9. Finally, the number of FSS unit cells selected is 7*7 to obtain a wider impedance bandwidth and a wider gain impedance.

The simulated reflection coefficient of the F-P antenna is shown in Figure 14. Its impedance bandwidth is slightly lower than that of the single antenna, and the relative bandwidth exceeds 20%. The simulated gain diagram of the feed antenna and F-P antenna is shown in Figure 15. It can be seen from the figure that the gain of the feed antenna is significantly improved in the 8 GHz~11.8 GHz frequency band. The relative gain bandwidth (3 dB bandwidth) of the F-P antenna is about 33%, which is greatly improved compared to the traditional F-P antenna. The efficiency of the antenna with and without the proposed PRS is simulated and provided in Figure 16. It can be found that the efficiency of the F-P antenna is lower than that of the individual antenna after 10.2 GHz but largely in line with the antenna before 10.2 GHz, presumably due to the cladding material.

## 3. Experimental Verification

In order to verify the performance of this F-P antenna, an F-P antenna prototype was fabricated, and its realized gain and radiation pattern were measured. The prototype is shown in Figure 17, and its measurement device in the anechoic chamber is shown in Figure 18. According to the test results in Figure 19, the working frequency range of the antenna is 8.3 GHz~11.25 GHz and the relative bandwidth is 29.5%, which basically agrees with the simulation result.

The measured and simulated gains of the prototype antenna are shown in Figure 20, together with the simulated gain of the feed antenna without PRS. As shown, the antenna gain could be significantly increased in the frequency range of 8.1 GHz~11.25 GHz, with a maximum in-band gain of 10.35 dBi and gain bandwidth of 31.5%. The differences between the measured and simulated results are due to antenna fabrication, assembly errors, and the actual tolerances of the measurement system. In addition, the gain comparison technique used to measure antenna gain may introduce some errors in the gain measurement.

As shown in Figure 20, the antenna gain was higher than that of a single antenna in the actual experiment, but it is different from the simulation. It is assumed that the cavity height *h* has a great influence on the experiment and that there is a large error in the actual manufacturing and simulation of *h*. We changed the cavity height *h* and used HFSS15.0 for modeling and simulation, and the results are shown in Figure 21. It can be seen that when *h* = 16.5 mm, the simulated gain curve is close to the actual measurement, so *h* has a great impact on the performance of the F-P antenna, and the experimental results are different due to the influence of *h*.

Another important characteristic of the antenna is the radiation, which was simulated and measured on the E-plane (*yoz*) and H-plane (*xoz*) of 9.5 GHz, 10 GHz, and 10.5 GHz, as shown in Figure 22. The F-P antenna produces directive radiation in both the E-plane and the H-plane with peak sidelobes less than −15 dB and broadside radiation. The high radiation intensity at the backside is caused by the back radiation from the feed antenna and SMA connectors.

A performance comparison of the proposed F-P resonator antenna with previous works is shown in Table 3. The work in [11] achieved a higher gain bandwidth and larger gain than the existing work, but [11] used an open waveguide as the main feed antenna, which was bulky and increased the structural cost. Compared with the listed literature, the antenna proposed in this paper has a higher gain, with the maximum reaching 10.35 dBi; a lower overall profile, about 0.6λ; and a wider gain bandwidth, with the relative gain bandwidth reaching 31.5%.

## 4. Conclusions

In this paper, a Fabry–Pérot resonator antenna with a double FSS as the cladding of a slit-coupled microstrip antenna is presented. In general, the proposed F-P antenna achieves high gain and broadband performance. The proposed FSS consists of a pure dielectric layer and a double-layered FSS structure. By studying the performance of the double-layered structure, the optimal parameters were obtained, and the equivalent circuit model was obtained. Finally, it has a positive reflection gradient phase at 9–11 GHz, which can effectively increase antenna gain and expand antenna bandwidth. According to the working principle of the Fabry–Pérot resonator, a wideband gap coupling is proposed, and the designed frequency-selective surface is combined with the microstrip antenna. By studying the cavity height *h* and number of FSS unit cells of the F-P antenna, *h* = 14.5 mm, and 7*7 FSS unit cells were selected to broaden the impedance bandwidth and the gain bandwidth simultaneously. The experimental results show that the impedance bandwidth of the F-P antenna is 29.5%, gain bandwidth is 31.5%, and maximum gain is 10.35 dBi, all higher than those of a single antenna. The proposed broadband F-P resonator antenna has potential application value in x-band radar and tracking systems.

## Figures and Tables

**Figure 1 sensors-22-09698-f001:**
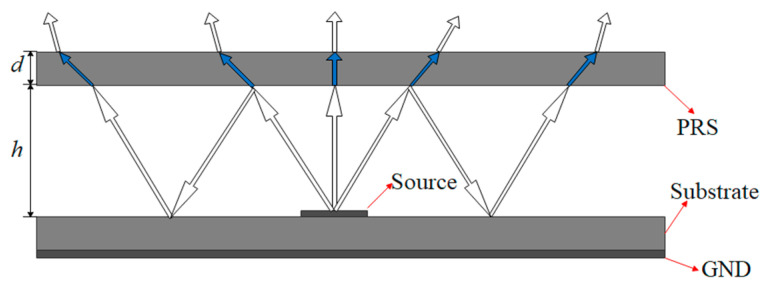
Fabry–Pérot resonant cavity antenna.

**Figure 2 sensors-22-09698-f002:**
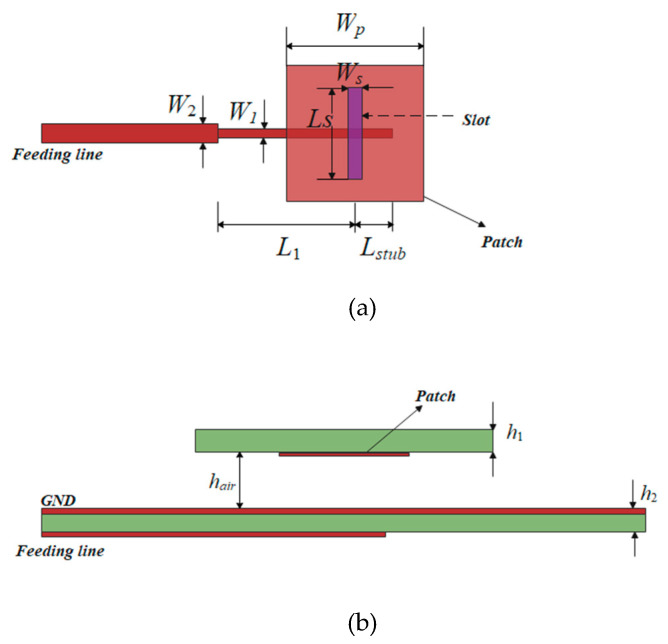
Structure of the microstrip slot antenna. (**a**). Top view. (**b**). Front view.

**Figure 3 sensors-22-09698-f003:**
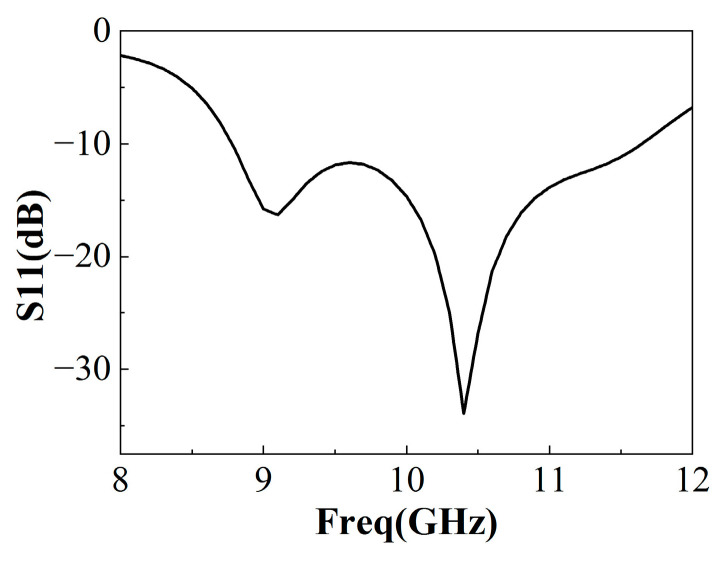
Reflection coefficient of feed antenna.

**Figure 4 sensors-22-09698-f004:**
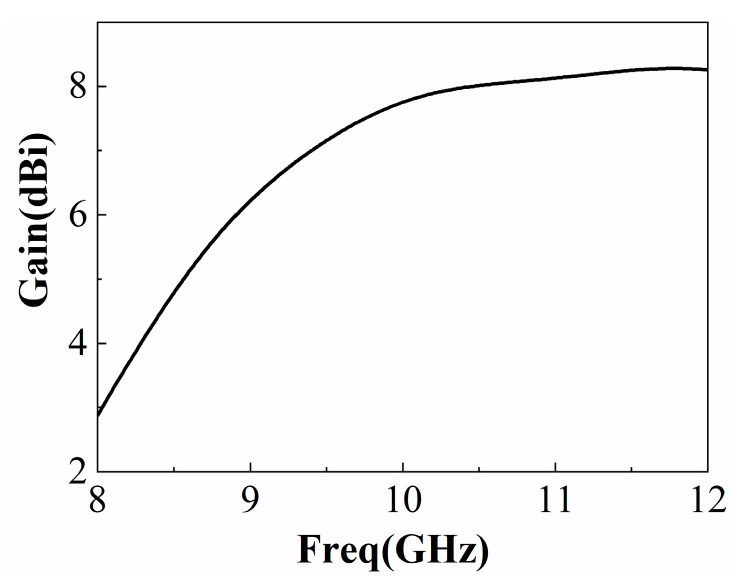
Reflection coefficient and gain of feed antenna.

**Figure 5 sensors-22-09698-f005:**
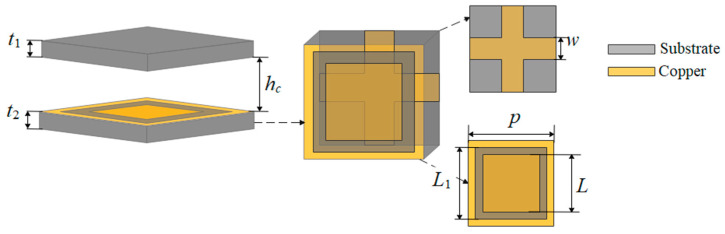
Overall structure of FSS unit.

**Figure 6 sensors-22-09698-f006:**
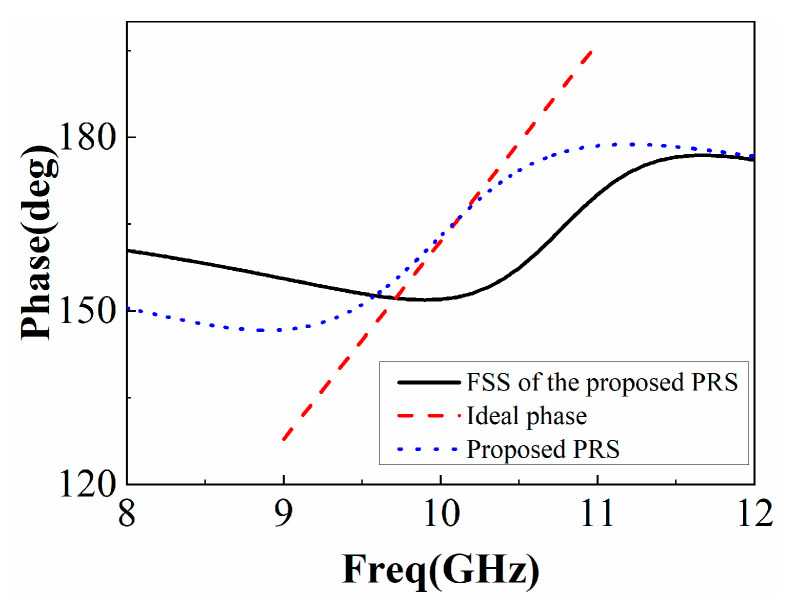
PRS reflection phase diagram.

**Figure 7 sensors-22-09698-f007:**
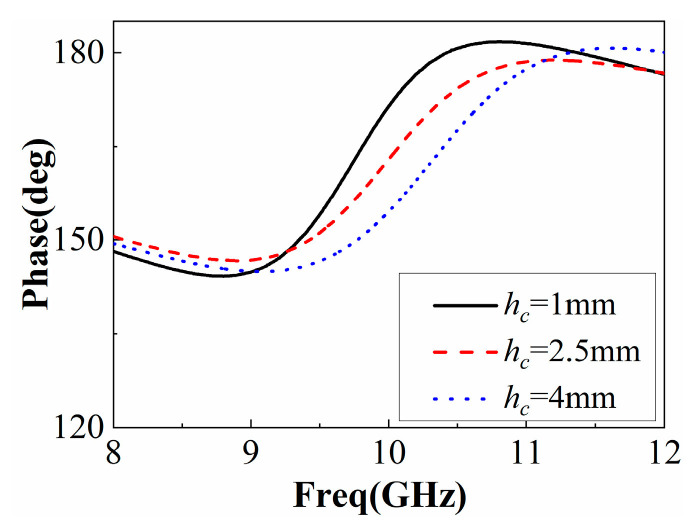
Variation of PRS reflection phase with *h_c_*.

**Figure 8 sensors-22-09698-f008:**
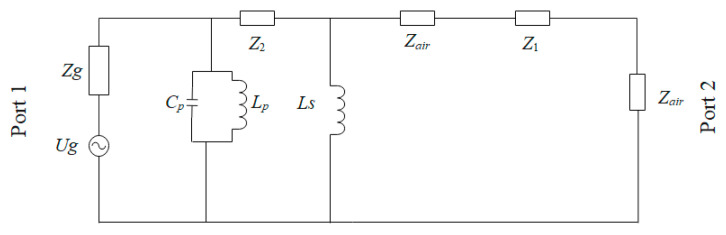
The equivalent circuit model (ECM) of the proposed FSS unit cell.

**Figure 9 sensors-22-09698-f009:**
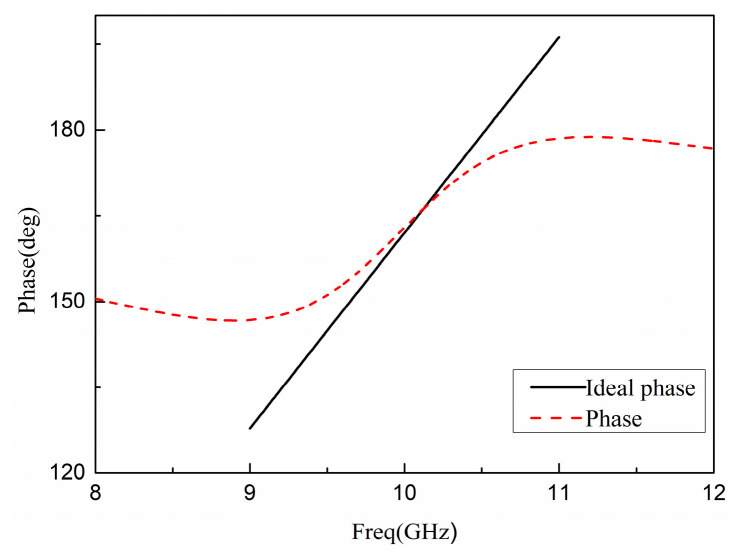
Phase of reflection coefficient.

**Figure 10 sensors-22-09698-f010:**
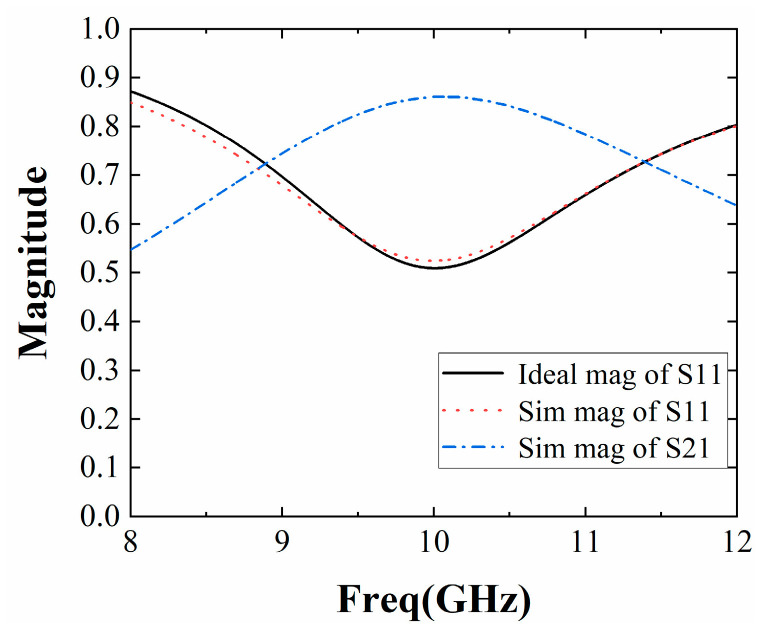
Amplitude of reflection coefficient and transmission coefficient.

**Figure 11 sensors-22-09698-f011:**
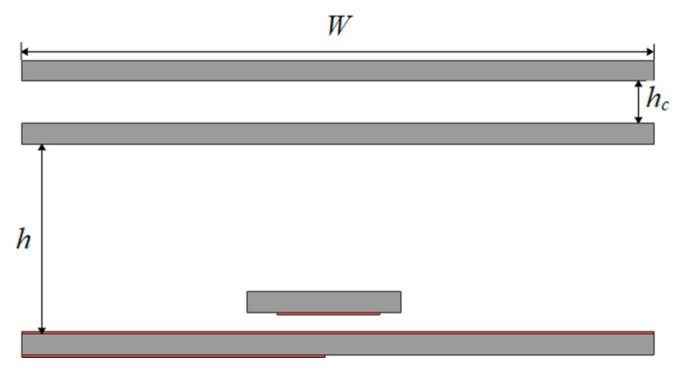
Schematic diagram of F-P resonant cavity antenna.

**Figure 12 sensors-22-09698-f012:**
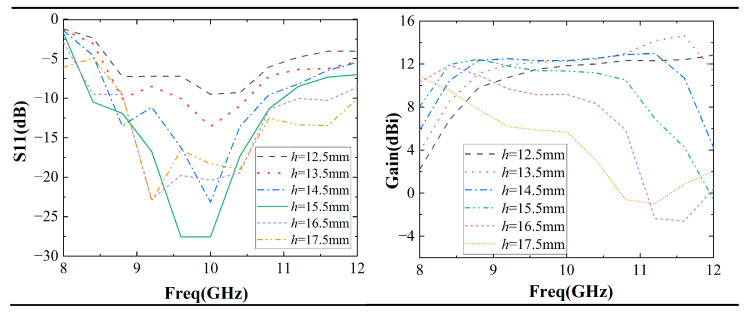
Simulation of F-P resonator antenna S11 and gain as a function of *h*.

**Figure 13 sensors-22-09698-f013:**
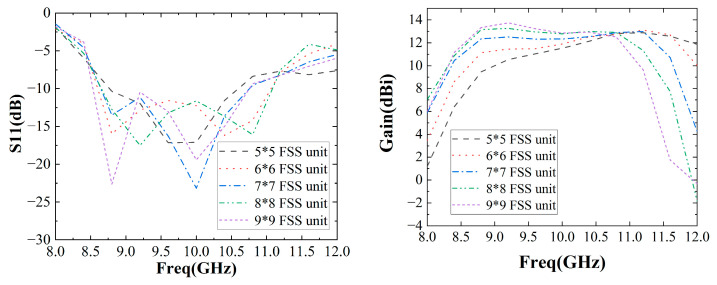
Simulation of F-P antenna S11 and gain as a function of number of FSS unit cells.

**Figure 14 sensors-22-09698-f014:**
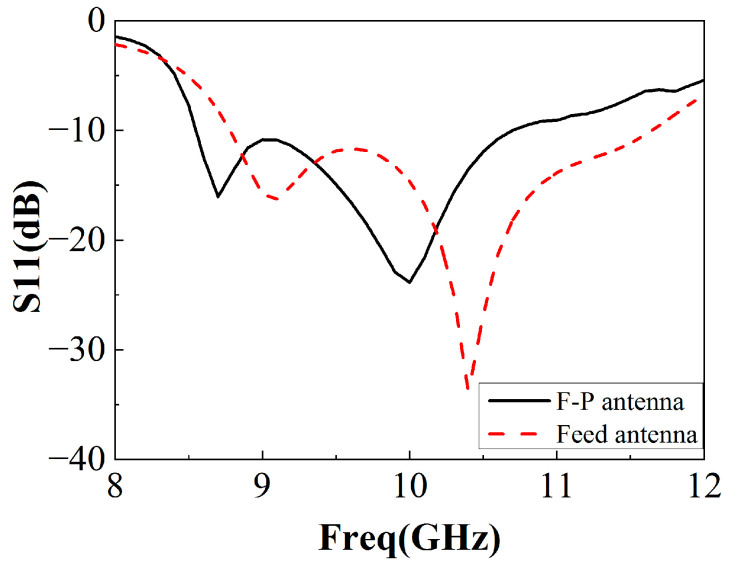
Reflection coefficient of F-P antenna and feed antenna.

**Figure 15 sensors-22-09698-f015:**
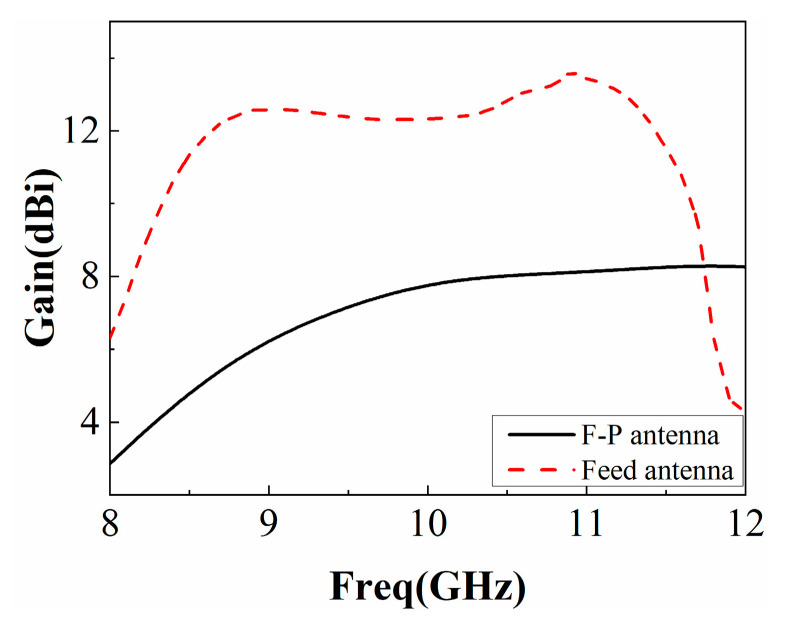
Gain of F-P antenna and feed antenna.

**Figure 16 sensors-22-09698-f016:**
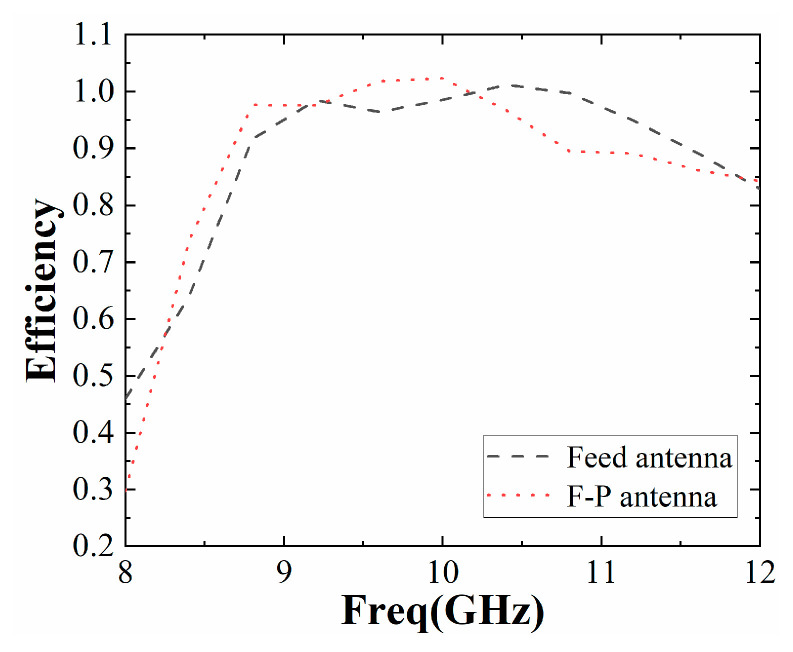
Efficiency of antenna.

**Figure 17 sensors-22-09698-f017:**
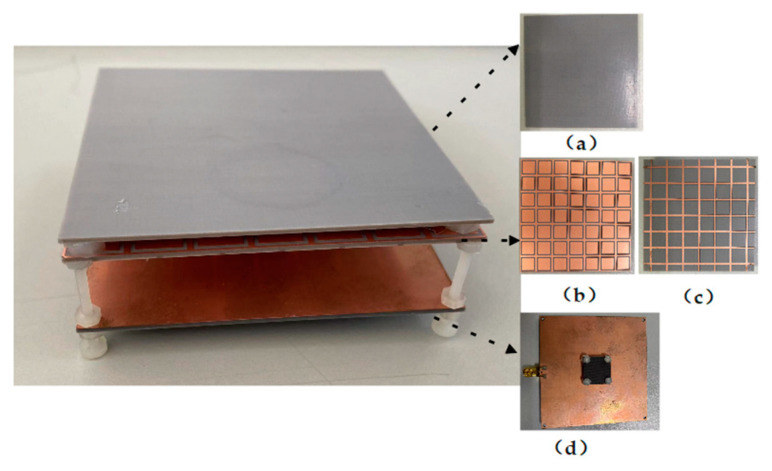
F-P antenna physical diagram. (**a**) PRS top layer. (**b**) PRS bottom layer upper surface. (**c**) PRS bottom layer lower surface. (**d**) Feed antenna.

**Figure 18 sensors-22-09698-f018:**
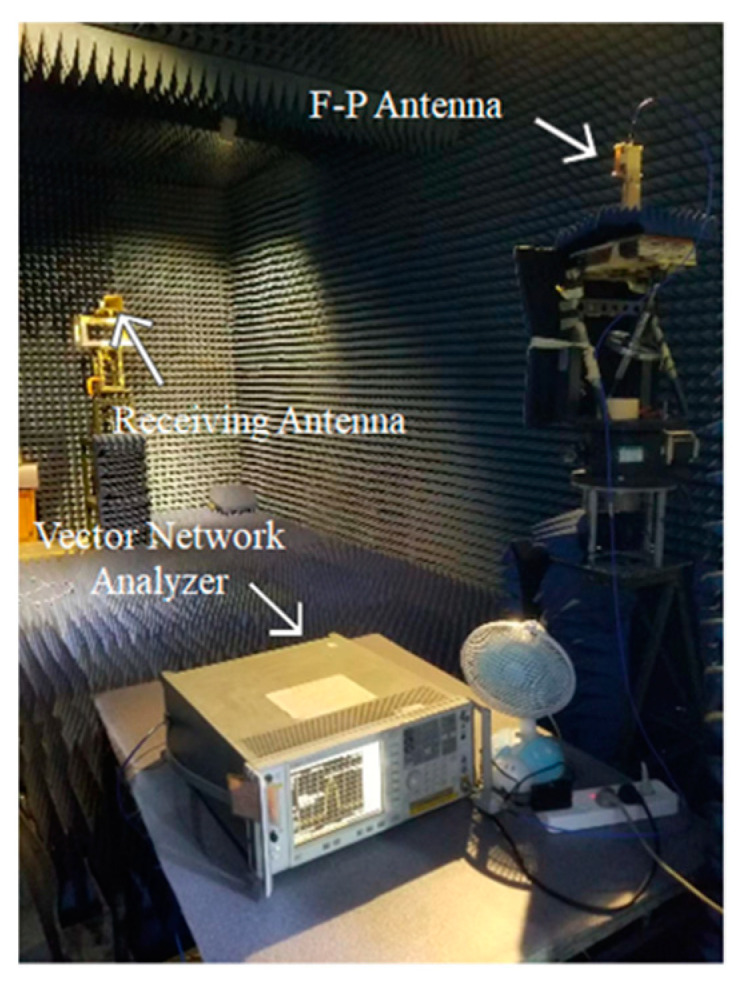
Testing diagram.

**Figure 19 sensors-22-09698-f019:**
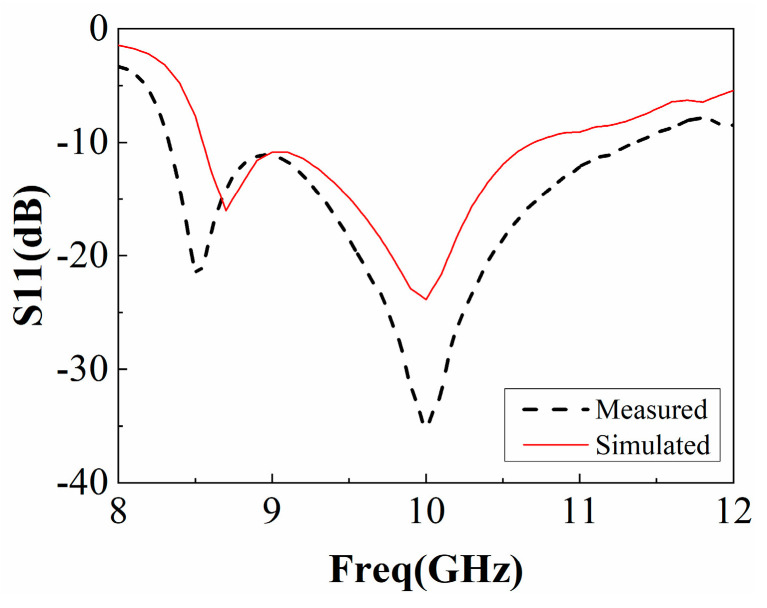
Measurement and simulation of F-P antenna S11.

**Figure 20 sensors-22-09698-f020:**
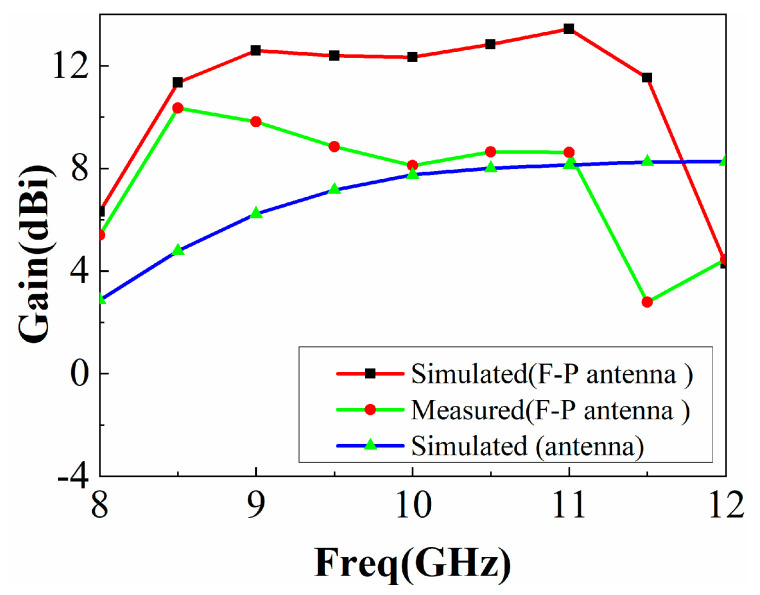
Prototype antenna’s gain measurement and simulation.

**Figure 21 sensors-22-09698-f021:**
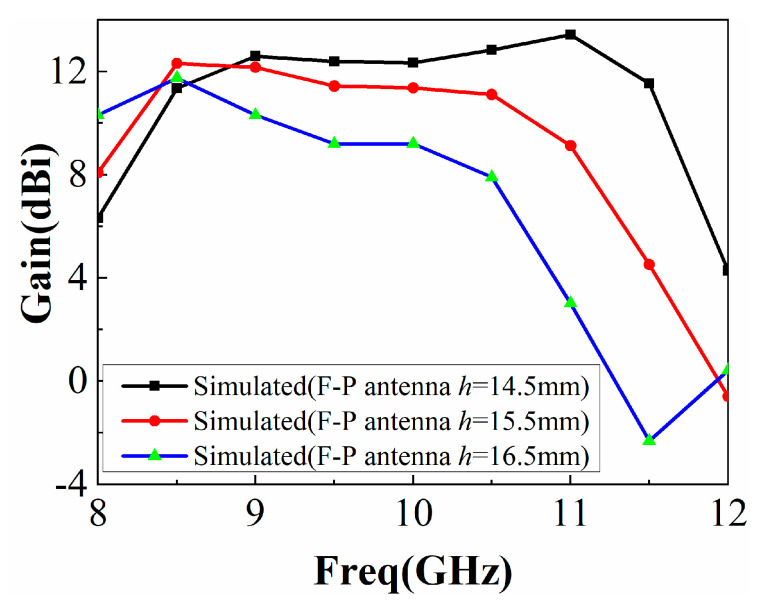
Simulation of F-P resonator antenna gain as a function of *h*.

**Figure 22 sensors-22-09698-f022:**
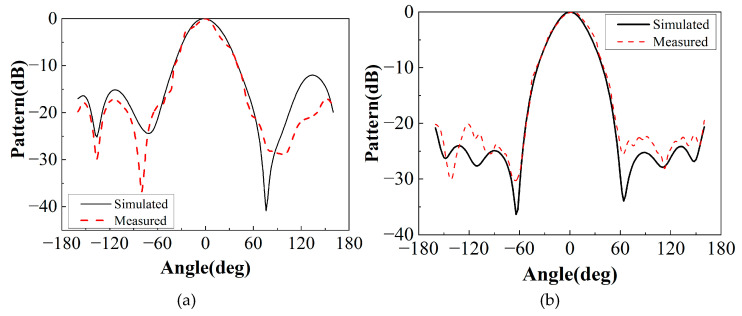
Radiation pattern of E and H surfaces of F-P antenna. (**a**) E-plane at 9.5 GHz; (**b**) 9.5 GHz H-plane. (**c**) E-plane at 10 GHz. (**d**) H-plane at 10 GHz. (**e**) E-plane at 10.5 GHz. (**f**) H-plane at 10.5 GHz.

**Table 1 sensors-22-09698-t001:** Structural parameters of microstrip slot antenna.

Parameters	Values	Parameters	Values
*W* _1_	1.2 mm	*W_s_*	1 mm
*W* _2_	2.3 mm	*L* _1_	9.5 mm
*W_P_*	8.8 mm	*L_stub_*	3 mm
*h_air_*	3 mm	*h* _1_	1 mm
*L_s_*	8.8 mm	*h* _2_	1 mm

**Table 2 sensors-22-09698-t002:** PRS structure parameters.

Parameters	Values
*P*	10 mm
*L* _1_	9 mm
*L*	7.5 mm
*W*	1 mm
*t* _1_	1 mm
*t_2_*	1 mm
*h_c_*	2.5 mm

**Table 3 sensors-22-09698-t003:** Performance comparison of broadband high-gain F-P antennas.

Ref	Freq (GHz)	Impedance BW S11 < −10 dB (%)	3 dB Gain BW (%)	Profile (λ0)	Primary Feed Antenna
[20]	7.45	8.69%	10.9%	0.64	Probe feed patch
[21]	12.5	16.2%	12.5%	1.4	Aperture-coupled patch
[22]	13	37%	32.3%	0.66	Waveguide feed
[23]	5.85	28.6%	22.3%	0.45	Probe feed with air Dielectric patch
[24]	13.8	12%	17.1%	0.65	Slot antenna
[25]	10.8	24.8%	15.9%	1	RIS-backed probe Feed patch
This paper	10	29.5%	31.5%	0.63	Air-loaded patch antenna

## Data Availability

Not applicable.

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
