# Peer review of "Design of Broadband High-Gain Fabry–Pérot Antenna Using Frequency-Selective Surface"

_sensors, 2022, doi:10.3390/s22249698_

Round 1

Reviewer 1 Report

The authors demonstrated the Design of a Broadband High Gain Fabry- Pérot Antenna using Frequency Selective Surface. The concept is exciting, and the simulation results are reasonably good, showing potentially strong reconfigurability. The antenna gain has been significantly improved when loaded with the FSS reflector. I have the following suggestions before accepting it for publication:

Introduction 

- The Introduction needs to be improved. The authors must explain some frequency selective surfaces (FSSs) reflectors and study their parameters, such as gain and bandwidth [1-3]. Please see these references, which may add value to the Introduction:

[1] Compact Size and High Gain of CPW-Fed UWB Strawberry Artistic Shaped Printed Monopole Antennas Using FSS Single Layer Reflector," in IEEE Access, vol. 8, pp. 92697-92707, 2020, doi: 10.1109/ACCESS.2020.2995069.

[2] High-Gain Wideband Circularly Polarised Fabry–Perot Resonator Array Antenna Using a Single-Layered Pixelated PRS for Millimetre-Wave Applications. Micromachines 2022, 13, 1658. https://doi.org/10.3390/mi13101658.

[3] A Wideband High-Gain Microstrip Array Antenna Integrated with Frequency-Selective Surface for Sub-6 GHz 5G Applications. Micromachines 2022, 13, 1215. https://doi.org/10.3390/mi13081215.

 2.2. Selection of feed antenna

- Please provide the magnitude of "h." parameter in mm.

-The FSS dual layer unit cells are too basic, which is a square loop and a four-legged loaded. We should support the contributions of this paper by studying the parametric study of the proposed FSS square loop and a four-legged loaded. In other words, the gap between the dual-band antenna and the FSS reflector "h" should also be studied and investigated. However, this parameter plays a crucial role in antenna engineering gain enhancement. Please create a parametric study based on the "h" parameter and show its effect on the S-parameter and Gain, respectively.

The Gap parameter "h" values should be as follows:

1- h= 12.5mm.

2- h=13.5mm.

3- h=14.5mm.

4- h=15.5mm.

5- h =16.5 mm.

6- h=17.5 mm.

  -Moreover, the number of FSS unit cells to structure the reflector is 7*7 unit cells. Another parametric study should be conducted and named the "Amount of FSS cells." parameter and show its effect on the S-parameter and Gain, respectively.

The "Amount of FSS cells" parameter values should be as follows:

1- 5*5.

2-6*6.

3-7*7.

4-8*8.

5-9*9.

To make it again clear, please add two graphs as a parametric study of the "Amount of FSS cells" parameter, as mentioned earlier.

To make it straightforward, please add two graphs as a parametric study of the "h" parameter in terms of S11 and gain.

2.3. Design of positive gradient reflection FSS unit

-Please provide the transmission coefficient (S21) graph of the dual-layer FSS unit cells.

-If possible, Please provide the equivalent circuit model (ECM) of the proposed FSS unit cell.

 2.4. Design of F-P resonant cavity antenna

-There needs to be a better understanding for me in Figure 13. The author's designed an FSS unit cell based on a square loop and a four-legged loaded for the upper and bottom one, as presented in Figure 5. But, in Figure 13, it seems like the other side of FSS is a grid-type aperture!!!! Please clarify.

Conclusions

The Conclusions should be rewritten with the updated results above.

-That's all for me at this moment. However, the authors are required to revise the comments above carefully!

Author Response

Thank you for your valuable comments, please refer to the attachment for specific answers.

Reviewer 2 Report

This paper presents a design of Fabry-Perot antenna using FSS structure. The paper is written well. There are some comments as follows.

1. Please add description of the reason why positive gradient reflection can improve the antenna bandwidth.

2. Add the definition of reflection gradient phase in page 5.

3. Please provide information of antenna efficiency ? Even small loss in substrate could lead to degraded efficiency in the febry-perot structure.

Author Response

(The authors gave the same response as above.)

Round 2

Reviewer 1 Report

The authors have revised the given comments successfully. However, there are still typos and spacing errors that need to be carefully checked.  

Best regards,

Author Response

Thank you for your effective advice. I have changed the article accordingly. Please see the attachment.
